# Quality of Life in Patients with Acute Severe Ulcerative Colitis: Long-Term Follow-Up Results from the CONSTRUCT Trial

**DOI:** 10.3390/jpm12122039

**Published:** 2022-12-09

**Authors:** Laith Alrubaiy, Hayley A. Hutchings, Andrea Louca, Frances Rapport, Alan Watkins, Shaji Sebastian, John G. Williams

**Affiliations:** 1St Mark’s Hospital, Watford Road, London HA1 3UJ, UK; 2Swansea University Medical School, Institute of Life Sciences 2, Swansea University, Singleton Park, Swansea SA2 8PP, UK; 3Australian Institute of Health Innovation, Centre for Healthcare Resilience and Implementation Science, Macquarie University, Sydney 2109, Australia; 4IBD Unit, Department of Gastroenterology, Hull University Teaching Hospitals NHS Trust, Anlaby Road, Hull HU3 2JZ, UK

**Keywords:** ulcerative colitis (UC), infliximab, ciclosporin, quality of life (QoL), EQ5D, CUCQ

## Abstract

Background: there is currently limited research examining the QoL of patients with Ulcerative colitis (UC) following treatment of acute severe colitis (ASUC). Objective: to examine the long-term QoL of ASUC patients enrolled in the CONSTRUCT trial following treatment of UC with infliximab or ciclosporin and to compare the differences in the QoL between the two drug treatments over time. Methods: The CONSTRUCT trial examined the cost and clinical effectiveness of infliximab and ciclosporin treatments for acute severe UC. We collected QoL questionnaire data from patients during the active trial period up to 36 months. Following trial completion, we contacted patients postannually for up to a maximum of 84 months. We collected QoL data using a disease-specific (CUCQ, or CUCQ+ for patients who had colectomy surgery) or generic (EQ5D-3L) questionnaire. We analysed QoL scores to determine if there was any difference over time and between treatments in generic or disease-specific QoL. Results: Following initial treatment with infliximab and ciclosporin, patients experienced a statistically significant improvement in both the generic and disease-specific QoL at three months. Generic scores remained fairly static for the whole follow-up period, reducing only slightly up to 84 months. Disease-specific scores showed a much sharper improvement up to 2 years with a gradual reduction in QoL up to 84 months. Generic and disease-specific QoL remained higher than baseline values. There was no significant difference between treatments in any of the QoL scores. Conclusions: Both infliximab and ciclosporin improve QoL following initial treatment for ASUC. QoL scores remain higher than at admission up to 84 months post-treatment.

## 1. Introduction

### Key messages

What is already known on this topic:Acute severe Ulcerative colitis (ASUC) can have a negative effect on patient quality of life;Drug treatment of ASUC can improve patient-reported quality of life;QoL should be assessed alongside clinical parameters.

What this study adds:Generic and disease-specific QoL improvements following drug treatment for ASUC are maintained long term (up to 84 months) after treatment;There are no differences in long-term QoL in patients receiving infliximab or ciclosporin as a rescue therapy for ASUC.

How this study might affect research, practice or policy:A colectomy did not appear to adversely affect QoL in patients postsurgery;It would be interesting to explore differences in QoL between patients with and without a stoma at the various time points and other factors that can affect QoL.

Ulcerative colitis (UC) is a chronic debilitating inflammatory bowel disorder [1]. The condition presents with a multitude of symptoms, predominantly bloody diarrhoea. The condition is characterised by an unpredictable clinical course with periods of exacerbation and remission [1]. There is an increasing prevalence of IBD across the world, with rates increasing both in the West and the East [1]. Increased rates across the world have been linked to lifestyle and environmental factors. UC incidence is bimodal, with the main age peak between the ages of 15 and 30 years, with a second smaller peak between the ages of 50 and 70 years [2].

The chronic and unpredictable nature of UC and its symptoms is associated with impaired patient quality of life (QoL) [3,4]. There may also be significant variation in symptoms both over time within the same patient and with different patients [5]. Measuring QoL in UC patients is important in order to assess changes in patients’ condition over time and following treatment, and also to gain an insight into patient perceptions of their condition and how this compares with clinical or objective outcomes [5]. Clinicians are now recognising the benefit of collecting QoL data from patients in addition to measuring clinical outcomes, and this has resulted in an increased collection of QoL data in routine clinical practice as well as being a key outcome in research trials of treatments [6].

Acute severe Ulcerative colitis (ASUC) affects up to 25% of patients with UC in their lifetime. The management of ASUC is based on early diagnosis and hospital admission for treatment with intravenous steroids [7]. Approximately 30% of hospitalised patients are resistant to steroid therapy and require rescue therapy with infliximab, ciclosporin [8,9] or colectomy [10,11]. In the last decade, new medical therapies have become available. Although these molecules can induce a rapid remission in moderate-to-severe colitis, their use in the management of ASUC is still limited [12]. 

We have previously reported the findings from the CONSTRUCT trial, which compared the clinical and cost effectiveness of infliximab and ciclosporin for steroid-resistant ASUC [1,2,3,4,5,6,7,8,10,11,12,13,14] and examined patients’ and healthcare professionals’ perception of disease course, experience and impact [15,16]. Although long-term clinical outcomes following treatment with infliximab or ciclosporin have been examined [9], we are not aware of any research that has explored the long-term QoL outcomes in patients receiving these treatments. This paper reports on the evaluation of long-term QoL in this group of ASUC patients for up to 84 months following treatment.

## 2. Materials and Methods

Two hundred and seventy patients were recruited to the CONSTRUCT trial [8,14], 135 being randomised to receive infliximab and 135 to receive ciclosporin. Demographic and clinical data were collected at the baseline, and clinical outcome data were recorded up to 36 months for all patients. In addition, patients completed questionnaires at baseline and at various time points throughout the trial (baseline, 3, 6, 12, 18, 24, 30 and 36 months) up to 36 months. 

Following the end of the trial, the patients were sent QoL questionnaires (EQ-5D and CUCQ/CUCQ+) annually for a further 48 months (time from baseline to 48, 60, 72 and 84 months) with the maximum follow-up period being 84 months. Reminder letters were sent to patients after four weeks if they were not initially returned. If the questionnaires were not returned after the second reminder, no further contact was made.

### 2.1. QoL Questionnaires

#### 2.1.1. Crohn’s and Ulcerative Colitis Questionnaire (CUCQ) and CUCQ+ 

We used a new disease-specific inflammatory bowel disease questionnaire (Crohn’s and Ulcerative Colitis Questionnaire—CUCQ) which included additional questions (CUCQ+) when patients had a stoma following a colectomy [5,8]. The CUCQ, (formerly the CCQ) has been validated for patients with Crohn’s disease and UC [5,8,17]. We asked patients to complete this questionnaire at each follow-up time point following enrolment. If a patient underwent a colectomy procedure, they were asked to ignore questions on the CUCQ relating to bowel function and instead complete supplementary stoma questions (CUCQ+) [8]. The CUCQ is made up of 32 items. Of these questions, there were six (Q1, Q2, Q6, Q9, Q24 and Q26) that were not relevant to postcolectomy patients [9]. We therefore designed the CUCQ+ with 10 stoma-specific questions replacing these six questions, making a total of 36 questions [2].

To calculate the scores for participants who had not undergone surgery, we used the 32 CUCQ questions. For postcolectomy participants, we used the 10 stoma-specific questions and the 26 stoma-relevant CUCQ questions to calculate the CUCQ+ scores. When analysing both CUCQ and CUCQ+, we calculated the scores as previously published [5,8,17]. The scores were only calculated where patients had completed enough questions to warrant a calculation of the CUCQ or CUCQ+ scores. Where missing data exceeded the recommended levels, patients were allocated a missing total score.

#### 2.1.2. EQ-5D-3 Level (EQ5D-3L)

We used the EQ5D-3 (https://euroqol.org/eq-5d-instruments/eq-5d-3l-about/ accessed 1st July 2010) to measure generic quality of life. The EQ5D-3L is made up of five general health questions, with three response options as well as a visual analogue scale (VAS) thermometer which asks patients to score their health on a scale from 0 to 100 [18]. We calculated a score for both the VAS (EQ5D VAS) and the questionnaire responses (EQ5D score). We scored the EQ5D according to the published guidance [18].

### 2.2. Analysis

We calculated CUCQ/CUCQ+ and EQ5D scores at each time point. We used a two-way analysis of variance to determine whether there were any differences over time and between treatments with the disease-specific (CUCQ or CUCQ+) or generic (EQ-5D scores and VAS) QoL measures. A Chi-squared test was used to compare the colectomy rate between treatments in the extended follow-up period. A *p* value of less than 0.05 was regarded as significant. We undertook the analysis using SPSS version 26 (IBM) licensed for Swansea University, UK.

### 2.3. Ethical Considerations

The CONSTRUCT protocol [13], patient information sheets and consent forms were approved by the Research Ethics Committee for Wales in July 2008 (Ref 08/MRE09/42) and subsequently by local UK National Health Service (NHS) research and development committees. The study had EudraCT Number (2008-001968-36) and clinical trial authorisation from the MHRA. The study conformed to the ethical guidelines of the 1975 Declaration of Helsinki as reflected in a priori approval by the Research Ethics Committee for Wales. 

All patients gave written informed consent to participate in the trial, including consent to receive further questionnaires for up to 10 years following trial enrolment and for their trial data to be linked to routinely collected data held in central returns. We undertook analysis on anonymous data having removed all identifiable information, thereby maintaining confidentiality.

## 3. Results

Table 1 illustrates the baseline characteristics of the patients within the infliximab and ciclosporin groups. There were no significant differences between groups at baseline with respect to any of the clinical characteristics. Table 2 and Figure 1 illustrate the generic and disease-specific quality of life scores at each time point in the infliximab and ciclosporin groups. Quality of life scores across all measures were low at baseline. Figure 1 illustrates that generic EQ5D VAS and EQ5D scores saw a rapid improvement up to 3 months, with a further gradual improvement up to 30 months. The scores then remained reasonably static for the remainder of the follow-up period. Disease-specific CUCQ scores showed a steep improvement up to 3 months, which continued to further improve up to 24 months. The CUCQ quality of life scores then started to gradually decline over time. Neither the generic nor the disease-specific quality of life scores fell back to the low levels documented at baseline.

ANOVA tests (Table 3) illustrated that there were no significant differences in any of the quality of life measures between the two treatment groups. There were, however, significant differences in quality of life scores over time. A posthoc analysis (Appendix A) identified that there were statistically significant differences between the EQ5D VAS scores between baseline and all time points up to 72 months. Despite the scores at 84 months being much higher than at baseline for the EQ5D VAS, there was no statistically significant difference. There were, however, only results for five patients at 84 months and substantial variability in their EQ5D VAS scores.

The EQ5D scores similarly showed a statistically significant improvement from baseline at all time points up to 84 months. There was also a statistically significant improvement between the 3-month scores and the 84-month scores. Although the EQ5D scores at 84 months were higher than at 3 months, there were only five patients at 84 months and substantial variability in the scores.

The CUCQ scores showed a statistically significant improvement from baseline at all time points up to 84 months. There was also a significant improvement in scores between 3 months and 24 months. There was a statistically significant deterioration in CUCQ scores between 18 months and 36, 48, 60 and 72 months. The scores at 24 months were also significantly lower than those at the 36-, 48-, 60- and 72-month time points. 

The number of patients documenting that they had a stoma in the follow-up period was not significantly different (*p* = 0.855) between the infliximab (29/102, 28.43%) and the ciclosporin (27/99, 27.27%) groups.

## 4. Discussion

Long-term studies of QoL in UC patients have focussed on patient-reported outcomes while on treatment for mild-to-moderate UC [20] and general QoL following the UC disease course [21], with indications that UC patients fare very well compared to the general population when measured using generic QoL tools. To our knowledge, this is the first study that has examined long-term QoL in ASUC patients following treatment with infliximab and ciclosporin.

We have shown that both generic and disease-specific QoL improves following treatment with infliximab and ciclosporin. The improvements following the initial treatment are pronounced for both generic and disease-specific quality of life but appear to be more consistently maintained for generic QoL up to 84 months. The improvements in disease-specific QoL seen in the early years gradually reduced over time up to 84 months. However, even at 84 months, the QoL levels were higher than those seen at baseline. There was no statistically significant difference in generic or disease-specific QoL scores between the infliximab and ciclosporin groups. There was also no difference in the proportion of patients who had colectomy surgery between the two treatment groups in the extended follow-up period.

It is interesting to see that early improvements in generic quality of life were maintained more than the improvements seen in disease-specific QoL. The average age of our population at recruitment was around 40, so it is possible that their increasing age may have also affected their QoL. It may be that as patients age, they are more resigned to accept that their ability to undertake general activities is reduced and that they feel on the whole that their QoL is good. Indeed, the scores we saw on the EQ5D are comparable to normal population levels. In terms of the disease-specific quality of life, there was a sharp improvement following initial treatment, and although patient QoL was still higher than at baseline, the longer-term effects of the treatment did not persist.

CONSTRUCT [8,13] was the first randomised controlled trial (RCT) in inflammatory bowel disease to incorporate patient-recorded quality of life as part of the primary outcome measure. Our findings confirmed no significant difference in clinical outcomes between infliximab and ciclosporin, a result that mirrored the findings of Laharie et al. [22]. A systematic review published in 2016, which included these two RCTs, also found that there were no significant differences in clinical outcomes and adverse events between the two treatments [23]. Laharie et al. have since followed up with patients at 5 years to examine disease-free survival and colectomy rates [9]. Their follow-up findings confirmed their initial findings that the long-term clinical outcomes of UC patients do not differ significantly between infliximab and ciclosporin treatment [9]. UC is known to significantly impact QoL [3,4,24,25], but this was not measured in the long-term Laharie study [9]. The CONSTRUCT trial [8,14] but did not explore longer-term QoL outcomes in ASUC patients receiving infliximab or ciclosporin rescue therapy and whether these longer-term outcomes differed between the two groups. 

In the CONSTRUCT trial, the percentage of patients having a colectomy was 45% [8,14]. In our long-term follow-up analysis of those patients responding, the percentage of colectomies was comparable between groups (around 27% in each group). It is encouraging to see that despite the large number of colectomies, patient QoL did not appear to be adversely affected. This concurs with findings from other studies where surgery in UC patients did not result in adverse effects on QoL [26,27]. A recent study of 10-year long-term outcomes in UC patients following ileal pouch anal anastomosis similarly showed good QoL outcomes in most patients [28].

There are some limitations of this work. This paper focuses only on follow-up information about QoL that was provided by the patients. Further work is planned utilising anonymised routinely collected health data to examine the influence of other factors that may have impacted their health, for example, further surgery and other comorbidities. It would be interesting to explore differences in QoL between patients with and without a stoma at various time points. Due to the small numbers at each time point, a detailed exploration of this was not possible, but a future study to explore this is warranted. The number of patients responding at each time point also fell at each time point, with limited data available at 84 months. This is likely due to patients moving or dying within the follow-up period or a lack of enthusiasm to respond given that they are no longer ‘actively’ being monitored.

## Figures and Tables

**Figure 1 jpm-12-02039-f001:**
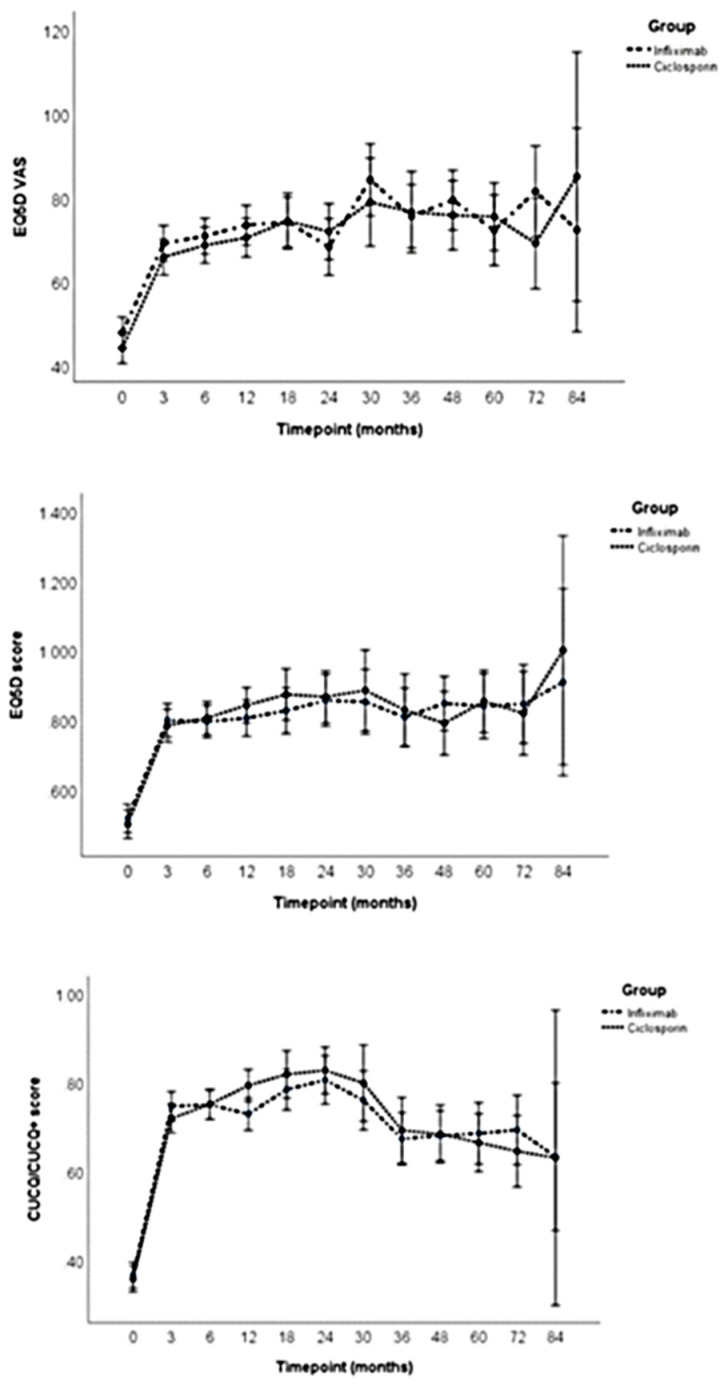
Mean (95% CI) quality of life profiles over time in the infliximab and ciclosporin groups for the EQ5D VAS, EQ5D score and CUCQ/CUCQ+.

**Table 1 jpm-12-02039-t001:** Baseline characteristics of recruited patients in the infliximab and ciclosporin treatment groups [13].

Variables	Infliximab (n = 135)	Ciclosporin (n = 135)
Age at randomisation (years), mean (SD) (n)	39.3 (15.5) (135)	39.8 (15.0) (135)
Gender: proportion, n (%)		
Female Male	46/135 (34.1) 89/135 (65.9)	54/135 (40.0) 81/135 (60.0)
Ethnicity: proportion, n (%)		
White Asian or Asian British Black or Black British Other ethnic groups	126/134 (94.0) 5/134 (3.7) 2/134 (1.5) 1/134 (0.7)	134/133 (63.2) 7/133 (5.3) 1/133 (0.8) 1/133 (0.8)
Weight (kg), mean (SD) (n)	74.3 (15.0) (135)	73.9 (15.3) (134)
Smoking: proportion, n (%)		
Never smoked/nonsmoker Current/ex-smoker	58/130 (44.6) 72/130 (55.4)	75/134 (56.0) 59/134 (44.0)
Family history: proportion, n (%)		
Yes (any one of mother, father, siblings, child) No	28/132 (21.2) 104/132 (78.8)	19/135 (14.1) 116/135 (85.9)
Condition severity (using TrueLove and Witts criteria [19]): proportion, n (%)		
Severe Not severe	97/133 (72.9) 36/133 (27.1)	95/131 (72.5) 36/131 (27.5)
Montreal score: proportion, n (%)		
E1 E2 E3	7/124 (5.6) 64/124 (51.6) 53/124 (42.7)	10/136 (7.9) 54/126 (42.8) 62/127 (49.2)
Mayo score, n (%)		
0 1 2 3	2/131 (1.5) 2/131 (1.5) 35/131 (26.7) 92/131 (70.2)	1/128 (0.8) 2/128 (1.6) 35/128 (27.3) 90/128 (70.3)
Receiving any of azathioprine, 6-mercaptopurine or methotrexate at baseline, n (%)		
At least one None	16/135 (11.9) 119 (135 (88.1)	26/135 (19.3) 69/135 (80.7)
Duration of symptoms for current episode (days), mean (SD) (n)	37.6 (46.0) (135)	41.4 (57.5) (131)

**Table 2 jpm-12-02039-t002:** Mean (SD) EQ5D VAS, EQ5D score and CUCQ/CUCQ+ scores for the infliximab and ciclosporin treatment groups.

Quality of Life Measure_Timepoint_Months	Infliximab			Ciclosporin		
	N	Mean	SD	N	Mean	SD
EQ5D_VAS_0	129	47.88	22.63	133	44.20	21.72
EQ5D_VAS_3	96	69.07	21.46	97	65.87	19.87
EQ5D_VAS_6	100	70.86	22.28	96	68.69	23.10
EQ5D_VAS_12	79	73.47	20.52	83	70.46	26.25
EQ5D_VAS_18	50	74.18	21.90	40	74.40	14.61
EQ5D_VAS_24	39	68.28	29.18	40	71.93	23.10
EQ5D_VAS_30	24	84.25	10.41	16	78.94	15.95
EQ5D_VAS_36	31	75.55	18.60	19	76.58	18.50
EQ5D_VAS_48	35	79.40	15.38	26	75.81	15.61
EQ5D_VAS_60	25	72.24	19.25	27	75.48	19.53
EQ5D_VAS_72	15	81.47	12.88	15	69.13	15.11
ED5D_VAS_84	3	72.33	25.33	2	85.00	7.07
EQ5D score_0	131	0.5172	0.2964	134	0.5000	0.3174
EQ5D score _3	95	0.7990	0.2121	98	0.7834	0.2354
EQ5D score _6	100	0.7954	0.2397	97	0.8051	0.2263
EQ5D score _12	80	0.8052	0.2259	83	0.8420	0.2279
EQ5D score _18	50	0.8265	0.2193	40	0.8732	0.1274
EQ5D score _24	39	0.8563	0.1815	39	0.8660	0.1850
EQ5D score _30	25	0.8522	0.1312	16	0.8846	0.1688
EQ5D score _36	31	0.8080	0.2218	20	0.8280	0.2333
EQ5D score _48	36	0.8473	0.1958	26	0.7905	0.2543
EQ5D score _60	25	0.8398	0.2000	27	0.8533	0.1913
EQ5D score _72	17	0.8456	0.1553	15	0.8192	0.1408
EQ5D score _84	3	0.9083	0.1588	2	1.0000	0.0000
CUCQ/CUCQ+_0	134	0.3666	0.1332	133	0.3574	0.1325
CUCQ/CUCQ+_3	99	0.7455	0.1830	103	0.7187	0.1855
CUCQ/CUCQ+_6	101	0.7497	0.1952	99	0.7505	0.2083
CUCQ/CUCQ+_12	82	0.7284	0.2110	86	0.7927	0.1738
CUCQ/CUCQ+_18	52	0.7837	0.1769	39	0.8179	0.1321
CUCQ/CUCQ+_24	37	0.8051	0.1706	40	0.8266	0.1255
CUCQ/CUCQ+_30	25	0.7592	0.1601	15	0.7981	0.1216
CUCQ/CUCQ+_36	32	0.6730	0.1531	20	0.6916	0.1739
CUCQ/CUCQ+_48	35	0.6797	0.1548	26	0.6835	0.1579
CUCQ/CUCQ+_60	23	0.6848	0.1672	26	0.6642	0.1498
CUCQ/CUCQ+_72	18	0.6928	0.1272	17	0.6447	0.1498
CUCQ/CUCQ+_84	4	0.6325	0.1401	1	0.6300	0

**Table 3 jpm-12-02039-t003:** Two-way ANOVA results comparing quality of life scores over time and between groups.

		F Value	Significance
EQ5D VAS	Group Time Group*Time	0.23 29.73 0.54	0.63 0.00 * 0.88
EQ5D score	Group Time Group*Time	0.26 33.85 0.35	0.61 0.00 * 0.97
CUCQ/CUCQ+	Group Time Group*Time	0.10 102.59 0.90	0.75 0.00 * 0.54

* *p* < 0.05.

## Data Availability

The data presented in this study are available on request from the corresponding author. The data are not publicly available due to patients confidentiality.

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
