# Peer review of "Quality of Life in Patients with Acute Severe Ulcerative Colitis: Long-Term Follow-Up Results from the CONSTRUCT Trial"

_jpm, 2022, doi:10.3390/jpm12122039_

Round 1

Reviewer 1 Report

minor spelling mistakes - line 46. (differences)

grammar mistake - line 49 (remove "be" or change to affected)

Overall manuscript adds one more piece of information beyond the usual clinical end points in the care of inflammatory bowel disease. Looking into long term quality of life after the treatments patients are subjected to is key to improving the multidisciplinary care delivered in this population. 

Overall the article is well written and well presented. minor spelling mistakes were noted. Certainly one of the strenghts of this manuscript is the well matched cohorts used to compare their analyses. It is not surprising that known accepted medical treatments for IBD including biologics do provide some benefit, as shown in the paper. Interesting to know both treatments were equally accepted by patients in terms of efficacy, and side effects. 

Author Response

Dear respected reviewer,

Thank you for your constructive comments. The highlighted spelling mistakes were corrected.

Best regards

Reviewer 2 Report

Below are my suggestions for authors;

In the introduction section, information about the drugs and drug forms used in the treatment of this disease can be given. Targeted systems used for such treatments are frequently encountered in the literature. Examples of market preparations used in treatment can be given. A general summary can be made. I suggest the authors review the following example.

 Impact of MMX® mesalamine on improvement and maintenance of health-related quality of life in patients with ulcerative colitis. Hodgkins P, Yen L, Yarlas A, Karlstadt R, Solomon D, Kane S. Inflamm Bowel Dis. 2013 Feb;19(2):386-96. doi: 10.1002/ibd.23022.

The place of infliximab and ciclosporin among all treatment options can be mentioned. Comparative discussion can be made with clinical results obtained from treated patients in the literature.

Author Response

Dear respected reviewer,

Thank you for your comment. The manuscript is focused on the care of patients with acute severe colitis (AUSC). We edited the section in the introduction to discuss the current practice in managing AUSC.

Best regards